# The Preventive Effects of Salubrinal against Pyrethroid-Induced Disruption of Adult Hippocampal Neurogenesis in Mice

**DOI:** 10.3390/ijms242115614

**Published:** 2023-10-26

**Authors:** Abigail C. Toltin, Abdelmadjid Belkadi, Laura M. Gamba, Muhammad M. Hossain

**Affiliations:** Department of Environmental Health Sciences, Robert Stempel College of Public Health and Social Work, Florida International University, Miami, FL 33199, USA

**Keywords:** Deltamethrin, salubrinal, caspase-12, ER stress, adult neurogenesis, hippocampus

## Abstract

Environmental factors, including pesticide exposure, have been identified as substantial contributors to neurodegeneration and cognitive impairments. Previously, we demonstrated that repeated exposure to deltamethrin induces endoplasmic reticulum (ER) stress, reduces hippocampal neurogenesis, and impairs cognition in adult mice. Here, we investigated the potential relationship between ER stress and hippocampal neurogenesis following exposure to deltamethrin, utilizing both pharmacological and genetic approaches. To investigate whether ER stress is associated with inhibition of neurogenesis, mice were given two intraperitoneal injections of eIf2α inhibitor salubrinal (1 mg/kg) at 24 h and 30 min prior to the oral administration of deltamethrin (3 mg/kg). Salubrinal prevented hippocampal ER stress, as indicated by decreased levels of C/EBP-homologous protein (CHOP) and transcription factor 4 (ATF4) and attenuated deltamethrin-induced reductions in BrdU-, Ki-67-, and DCX-positive cells in the dentate gyrus (DG) of the hippocampus. To further explore the relationship between ER stress and adult neurogenesis, we used caspase-12 knockout (KO) mice. The caspase-12 KO mice exhibited significant protection against deltamethrin-induced reduction of BrdU-, Ki-67-, and DCX-positive cells in the hippocampus. In addition, deltamethrin exposure led to a notable upregulation of CHOP and caspase-12 expression in a significant portion of BrdU- and Ki-67-positive cells in WT mice. Conversely, both salubrinal-treated mice and caspase-12 KO mice exhibited a considerably lower number of CHOP-positive cells in the hippocampus. Together, these findings suggest that exposure to the insecticide deltamethrin triggers ER stress-mediated suppression of adult hippocampal neurogenesis, which may subsequently contribute to learning and memory deficits in mice.

## 1. Introduction

Pyrethroids, one of the major groups of pesticides used globally, account for roughly 30% of the worldwide pesticide market [1]. They are extensively used in agricultural and residential environments for insect and pest control as they are considered relatively safer than organophosphate pesticides. Notably, urinary metabolites of pyrethroid insecticides have been frequently detected in children, adults, and even pregnant women, indicating widespread exposure among the general population [2,3,4,5]. Deltamethrin is one of the commonly used type-II pyrethroids that primarily induce neurotoxicity by delaying the closing of voltage-gated sodium channels [6,7]. In addition to ion channels, other potential cellular mechanisms such as endoplasmic reticulum (ER) stress, oxidative stress, and neuroinflammation have also been identified as contributors to pyrethroid neurotoxicity.

Cognitive impairment is an emerging problem in the general population, with an estimated 10–15% of US adults aged 45 and above experiencing some form of cognitive deficits that can manifest as difficulty with learning, memory, language, concentration, and problem-solving [8]. It can affect people of all ages and is caused by several factors, including environmental exposure to air pollution, heavy metals (lead and mercury), and pesticides [9,10,11,12]. Studies have shown that exposure to pesticides is linked to cognitive deficits, particularly in agricultural workers, their family members, and people living near pesticide-treated areas [12,13,14]. Recent studies in rural Ecuador and China have found that children exposed to pesticides showed delayed language development and lower scores on cognitive tests [15,16]. Importantly, exposure to pyrethroid has been shown to be associated with cognitive dysfunction [17].

Endoplasmic reticulum (ER) stress is a cellular response that occurs when the ER becomes overwhelmed by different factors, including environmental toxicants [18,19]. ER stress has emerged as a significant contributor to neurodegenerative diseases, such as Alzheimer’s, Parkinson’s, and Huntington’s disease [20,21,22]. Prolonged or excessive ER stress can perturb cellular homeostasis, resulting in the buildup of misfolded proteins, oxidative stress, and inflammation, ultimately culminating in neuronal dysfunction and cell death. Environmental toxicants, including heavy metals, pesticides, and air pollutants, have been implicated in inducing ER stress, and consistent activation of the ER stress can lead to the loss of neurons, causing neurodegeneration and cognitive impairment [20,23]. Previous work from our laboratory has demonstrated that exposure to deltamethrin insecticide caused ER stress in the hippocampus and learning deficits in mice [24]. Most recently, we reported that repeated exposure to deltamethrin insecticide causes ER stress, cell loss, and impairs hippocampal neurogenesis in mice [19,24,25].

Neurogenesis, the process of generating new neurons, plays a significant role in hippocampal-dependent learning and memory [26,27,28,29]. Studies have revealed that reduced neurogenesis is associated with deficits in learning and memory [30,31]. Furthermore, cognitive impairments can arise from conditions that hinder neurogenesis, such as chronic stress or neurodegenerative diseases [32,33,34,35,36]. Environmental toxicants have been shown to have detrimental effects on neurogenesis. Exposure to certain toxic substances, such as lead, methyl mercury, pesticides, and air pollutants can disrupt neurogenic niches and impair the proliferation and survival of neural stem cells [37,38,39,40,41,42,43]. In previous studies, we demonstrated that exposure to deltamethrin induces ER stress, diminishes hippocampal neurogenesis, and impairs cognition in adult mice [25,41]. However, the potential relationship between ER stress and hippocampal neurogenesis has not been elucidated.

In this study, we investigated the potential connection between ER stress and hippocampal neurogenesis after exposure to deltamethrin, employing both pharmacological and genetic methodologies. Pre-treatment of mice with salubrinal prevents ER stress and reduces deltamethrin-induced reduction of neurogenesis in the dentate gyrus (DG) of the hippocampus, as evidenced by significantly improved counts of BrdU-, Ki-67-, and DCX-positive cells. Caspase-12 deficient mice showed protection against deltamethrin-induced ER stress and no significant impairment of hippocampal neurogenesis. Taken together, these findings suggest that disruption of adult neurogenesis occurs via ER stress following deltamethrin exposure.

## 2. Results

### 2.1. Salubrinal Administration Effectively Prevents the Activation of the ER Stress in the Hippocampus of Adult Mice Induced by Exposure to Deltamethrin

First, we confirm whether the inhibition of eIF2α with salubrinal prevents the induction of ER stress caused by deltamethrin. We employed Western blot analysis to determine the levels of ER stress-related proteins CHOP and ATF4 after 48 h of exposure. Our results revealed a significant increase of 231% in CHOP protein (Figure 1A) and 161% in ATF4 protein (Figure 1B) in the hippocampi of mice treated with a single dose of 3 mg/kg of deltamethrin, indicating a substantial induction of ER stress compared to the control group. Remarkably, our data demonstrated that pre-treatment of mice with salubrinal effectively prevents the deltamethrin-induced activation of CHOP (Figure 1A) and ATF4 (Figure 1B) proteins in the hippocampus. These findings confirmed that salubrinal can prevent deltamethrin-induced ER stress in the hippocampi of adult mice.

### 2.2. Caspase-12 KO Mice Are Protected from Deltamethrin-Induced ER Stress

Next, we conducted an additional study using caspase-12 KO mice, as caspase-12 is known to be involved in the activation of ER stress. Both caspase-12 KO and WT mice were given a single dose of 3 mg/kg deltamethrin via oral gavage and were sacrificed 48 h later for assessment of ER stress. The results showed that deltamethrin significantly increased the levels of ER stress-related proteins, CHOP, and ATF4 in the hippocampus of WT mice. In contrast, caspase-12 KO mice treated with deltamethrin exhibited significantly reduced levels of CHOP and ATF4 proteins in the hippocampus when compared to deltamethrin-treated WT mice (Figure 2). Specifically, CHOP protein levels increased by 234% in WT mice and 125% in caspase-12 KO mice (Figure 2A), while ATF4 levels increased by 260% in the hippocampi of WT mice and 153% in caspase-12 KO mice (Figure 2B) when compared to control animals. These findings indicate that caspase-12-deficient mice are protected against deltamethrin-induced ER stress in the hippocampus.

### 2.3. Inhibition of ER Stress with Salubrinal Attenuates Deltamethrin-Induced Reduction of Hippocampal Neurogenesis in Adult Mice

In a previous study, we demonstrated that deltamethrin exposure triggers ER stress and decreases hippocampal neurogenesis in adult mice. Here, we investigated the possible relationship between ER stress and hippocampal neurogenesis following acute exposure to deltamethrin (3 mg/kg). Total number of BrdU-, Ki-67- and DCX-positive cells are presented in Appendix A. Our findings revealed that acute exposure to deltamethrin resulted in an approximately 2.27-fold decrease in BrdU-positive cells in the hippocampus compared to the control group (Figure 3A). Administration of salubrinal prior to deltamethrin exposure significantly counteracted this reduction in BrdU-positive cells. Animals treated with salubrinal exhibited a slight decrease (1.2-fold) in BrdU-positive cells compared to the control group. However, there was a significant 1.8-fold increase (*p* < 0.05) in BrdU-positive cells in salubrinal pre-treated animals compared to those exposed to deltamethrin alone (Figure 3A).

We also assessed cell proliferation using the Ki-67 marker, as shown in Figure 3B. Animals exposed to deltamethrin exhibited a significant (*p* < 0.05) 1.67-fold decrease in Ki-67-positive cells in the hippocampus compared to the control group (Figure 3B). However, when animals were pre-treated with salubrinal before deltamethrin exposure, the number of Ki-67-positive cells did not show a significant decrease compared to the control group (Figure 3B). Instead, there was a noteworthy (*p* < 0.05) 1.5-fold increase in Ki-67-positive cells compared to animals exposed to deltamethrin alone (Figure 3B).

Next, we investigated the population of immature neurons using the DCX marker as shown in Figure 3C. Our results revealed a significant reduction in DCX-positive cells following exposure to deltamethrin. Importantly, this reduction was significantly attenuated by the inhibition of ER stress with salubrinal, as observed in deltamethrin-treated animals. Animals exposed to deltamethrin displayed a notable (*p* < 0.05) 1.4-fold decrease in DCX-positive cells in the hippocampus compared to the control group (Figure 3C). However, when animals were pre-treated with salubrinal, the number of DCX-positive cells did not exhibit a significant decrease compared to the corn oil control (Figure 3C). Instead, there was a remarkable (*p* < 0.05) 1.3-fold increase in DCX-positive cells compared to animals exposed to deltamethrin alone (Figure 3C). These findings suggest that the observed reduction of hippocampal neurogenesis may be due to the induction of ER stress following deltamethrin exposure.

### 2.4. Caspase-12 KO Mice Are Protected from Deltamethrin-Induced Reduction of Hippocampal Neurogenesis in Adult Mice

To investigate whether global knockout of caspase-12 provides protection against the effects of deltamethrin on hippocampal neurogenesis, we administered a single oral dose of 3 mg/kg deltamethrin to both caspase-12 KO and WT mice via oral gavage. The mice were then sacrificed 48 h later to evaluate progenitor cell proliferation and after 7 days to assess DCX-positive immature neurons, as no changes were observed at 48 h in our previous study [25]. Total number of BrdU-, Ki-67- and DCX-positive cells are presented in Appendix A. Our findings revealed that WT mice treated with deltamethrin exhibited an average loss of 46% in BrdU+ cells, whereas global caspase-12 KO mice demonstrated only an average of 10% loss (Figure 4A,B) compared to the WT control group. Additionally, we assessed Ki-67+ cells in these animals, and the results showed that deltamethrin-treated WT mice exhibited an average loss of 56% of Ki-67-positive cells, while caspase-12 KO mice exhibited only a 19% loss compared to corn oil-treated WT animals (Figure 4C,D). Next, we examined the population of immature neurons using the DCX marker. Deltamethrin-treated animals showed a significant (60%) reduction of DCX-positive immature neurons in WT mice, whereas the same treatment resulted in a 26% loss in caspase-12 KO mice compared to corn oil-treated WT animals (Figure 4E,F). Consequently, caspase-12 deficiency significantly mitigated the deltamethrin-induced reduction in progenitor cell proliferation as well as immature neurons. These results further suggest that exposure to deltamethrin may impair hippocampal neurogenesis via ER stress.

### 2.5. Salubrinal Treatment Attenuates the Expression of CHOP in Hippocampal NPCs in Deltamethrin-Treated Mice

To determine whether neural progenitor cells (NPCs) undergo ER stress after deltamethrin exposure, we examined the expression of CHOP in BrdU- and Ki-67-positive cells using double immunofluorescence staining. Total number of CHOP-expressing BrdU- and Ki-67-positive cells are presented in Appendix A. Our results showed that exposure to deltamethrin significantly increased CHOP-expressing BrdU-positve cells in the SGZ of the DG compared to the control animals (Figure 5). Similarly, CHOP-expressing Ki-67-positive cells were also significantly increased in deltamethrin-exposed animals compared to the control group (Figure 6). Interestingly, mice treated with salubrinal showed a significant reduction in the number of CHOP-expressing BrdU- and Ki-67-positive cells compared to animals treated with deltamethrin alone (Figure 5 and Figure 6).

### 2.6. Salubrinal Attenuates the Expression of Caspase-12 in Hippocampal NPCs in Deltamethrin-Treated Mice

Here, we examined the expression of caspase-12 in BrdU- and Ki-67-positive cells via double immunofluorescence staining to ascertain whether NPCs undergo apoptosis following exposure to deltamethrin, as ER stress-mediated apoptotic death signaling can occur via the sequential activation of caspase-12 followed by caspase-3. Deltamethrin treatment resulted in a significant increase in the number of caspase-12-expressing BrdU-positive cells by 27% in the dentate gyrus of the hippocampus compared to the control animals (Figure 7). Similarly, there was a significant rise in caspase-12-expressing Ki-67+ cells by 40% in the deltamethrin-exposed animals when compared to the control group (Figure 8). Mice pre-treated with salubrinal showed a significant reduction in the number of caspase-12-expressing BrdU+ and Ki-67+ cells in the hippocampi of deltamethrin-treated mice (Appendix A), indicating that NPCs may undergo ER stress-induced apoptosis after deltamethrin exposure. 

### 2.7. Caspase-12 Knockout Mice Are Protected against Deltamethrin-Induced Induction of ER Stress in Hippocampal NPCs

To determine whether NPCs undergo ER stress in caspase-12-deficient mice following exposure to deltamethrin, we examined the expression of CHOP in BrdU- (Figure 9) and Ki-67- (Figure 10) positive cells in the hippocampus. Total number of CHOP-expressing BrdU- and Ki-67-positive cells are presented in Appendix A. Our findings revealed that deltamethrin-treated caspase-12 KO mice exhibited a significantly reduced number of CHOP-expressing BrdU- and Ki-67-positive cells when compared to the WT animals treated with deltamethrin (Figure 9 and Figure 10). These results indicate that exposure to deltamethrin induces ER stress in NPCs, leading to potential inhibition of cellular proliferation and a reduction in hippocampal neurogenesis in adult mice.

## 3. Discussion

Chronic and persistent ER stress has been linked to the pathogenesis of neurological diseases that lead to cognitive impairment. Both epidemiological and laboratory-based studies have demonstrated that environmental toxicants, including pesticides, can trigger ER stress and increase the risk of developing cognitive impairment and various neurological disorders [44]. The hippocampus is an important part of the brain that seems to be more susceptible to the effects of deltamethrin exposure in adult animals [19]. In our previous study, we found that exposure to the pyrethroid insecticide deltamethrin caused hippocampal ER stress, impaired neurogenesis, and resulted in cognitive impairment in adult mice [19,24,25,41]. However, the mechanistic link between ER stress and disruption of neurogenesis following exposure to deltamethrin has not been fully established. In this study, we investigated the role of ER stress in the impairment of hippocampal neurogenesis in mice following acute exposure to deltamethrin. Our data showed that the inhibition of ER stress with salubrinal attenuated deltamethrin-induced reductions in progenitor cell proliferation (BrdU+ and Ki-67+ cells) and immature neurons (DCX+ cells) in the hippocampus. Furthermore, caspase-12 KO mice demonstrated significant protection against induction of deltamethrin-induced ER stress and reductions in BrdU+, Ki-67+, and DCX+ cells in the hippocampus. These results provide evidence that deltamethrin exposure leads to a reduction in adult hippocampal neurogenesis by inducing ER stress.

Exposure to environmental toxicants, including insecticides, herbicides, and fungicides, has become a growing concern due to potential adverse effects on human health. Toxicants have been linked to a higher prevalence of chronic neurodegenerative diseases resulting in cognitive impairment [12,45]. The increased usage of pyrethroid pesticides in recent years has raised concerns about their impact on human health as evidenced by the frequent detection of these pesticides in human urine, blood, and breast milk [1,46,47,48,49,50,51]. Here, we found that acute single exposure to 3 mg/kg of deltamethrin produced detrimental effects on hippocampal neurogenesis in adult mice. This outcome is as robust, as we observed after repeated chronic exposure to deltamethrin [24,41], and it aligns with other reports indicating that exposure to organophosphates, paraquat, organochlorines, carbamates, and methyl mercury impairs hippocampal neurogenesis and results in cognitive deficits in rodents [44,52,53,54,55].

Chronic ER stress is a significant contributing factor to neurodegeneration and cognitive deficits. There is strong evidence that ER stress induction leads to reduced neuronal viability and synaptic plasticity in the hippocampus [56,57]. Additionally, both human post-mortem brain and animal models of Alzheimer’s disease (AD) consistently reveal an upregulation of ER stress markers, accompanied by a decrease in the number of neural stem cells (NSCs) and immature granule neurons in the hippocampus [58,59]. In our earlier studies, we observed that exposure to deltamethrin resulted in a significant increase in the ER stress-specific protein CHOP, caspase-12, and ATF4 in the hippocampus, which were accompanied by a decrease in the proliferation of NPCs (BrdU+ and Ki-67+ cells) and immature neurons (DCX+ cells). However, the direct causal relationship between ER stress and the disruption of neurogenesis has yet to be definitively established. To investigate the role of ER stress in the reduction of hippocampal neurogenesis induced by deltamethrin, mice were pre-treated with salubrinal to pharmacologically inhibit the ER stress pathway. Salubrinal is a specific inhibitor of eIF2α dephosphorylation which serves as an initial signaling molecule in ER stress [19,60,61]. By preventing this dephosphorylation, salubrinal helps to reduce the burden on the ER and restore cellular homeostasis. Data from this study revealed that inhibiting eIF2α with salubrinal not only mitigated the heightened levels of ATF4 and CHOP but also led to an increase in the numbers of BrdU+, Ki-67+, and DCX+ cells in the hippocampus. These results were further validated using a genetic approach, in which caspase-12-deficient mice exhibited protective effects against deltamethrin-induced ER stress and reduction of these cells. Caspase-12 is an ER-resident cysteine protease typically present in an inactive state within the ER, and it becomes activated in response to ER stress. Studies demonstrated that cells expressing caspase-12 are susceptible to ER stress, whereas cells lacking caspase-12 exhibit resistance to ER stress [62,63]. These findings provide strong evidence supporting the involvement of ER stress in the disruption of adult neurogenesis following exposure to deltamethrin.

During ER stress, cells trigger unfolded protein response (UPR) to restore ER functions via three ER–transmembrane sensors such as protein kinase RNA-like endoplasmic reticulum kinase (PERK), inositol-requiring enzyme 1 (IRE1), and transcription factor 6 (ATF6) [64] In this study, we focused on downstream signaling of PERK pathways and found that deltamethrin caused disruption of hippocampal neurogenesis via the activation of elF2α–ATF4–CHOP signaling pathway. Upon the activation of the UPR in ER, eIF2α undergoes phosphorylation. This event triggers the translational induction of ATF4, which subsequently activates CHOP, ultimately leading to the propagation of pro-apoptotic signals [64]. Dysregulation of this signaling cascade has been implicated in neurodegeneration and neurodegenerative diseases. Our data demonstrate that inhibiting eIF2α dephosphorylation with salubrinal prevents ATF4 and CHOP induction in the hippocampus and mitigates CHOP and caspase-12 expression in NPCs in deltamethrin-treated mice.

CHOP is known as a growth arrest and DNA damage-inducible gene 153 (GADD153) serves as a central mediator of cell death triggered via ER stress [65,66]. CHOP is expressed at minimal levels under physiological conditions, but it becomes upregulated during ER stress and plays a crucial role in cell cycle arrest and apoptosis [66,67]. In our study, we observed a notable increase in CHOP protein in BrdU+ and Ki-67+ cells, as well as in DCX+ cells in DG of deltamethrin treated mice. These elevated CHOP levels were significantly reduced when animals were pre-treated with salubrinal. To validate these findings, we conducted an additional experiment utilizing caspase-12 KO mice. Interestingly, the caspase-12 KO mice showed notable protection against the deltamethrin-induced upregulation of CHOP in BrdU+ and Ki-67+ cells in the hippocampus. In addition, the number of caspase-12-expressing BrdU+ and Ki-67+ cells were significantly increased in deltamethrin-treated WT mice. These findings indicate that NPCs undergo ER stress following exposure to deltamethrin, which potentially contributes to the reduction of hippocampal neurogenesis. These results are consistent with others showing ER stress-induced CHOP expression in different contexts [68,69]. Notably, a study involving mice with long-term obesity revealed ER stress and CHOP expression in DCX-expressing immature neurons, indicating a common mechanism of ER stress-related neurogenesis impairment [70]. Similarly, prolonged ER stress was associated with decreased neuronal survival and disrupted neurogenesis in the hippocampi of mice with spinal cord injuries [71]. The role of ER stress in neurogenesis was further confirmed in a study involving paraquat exposure, where ER stress-mediated reduction of NSCs (Ki-67+ and SOX2+ cells) was observed in the mouse hippocampus [69]. Taken together, these findings underscore the significance of CHOP-mediated ER stress as a key player in the regulation of neurogenesis in various stressful conditions, including environmental exposure to pyrethroid insecticides.

Exposure to the pyrethroid insecticide deltamethrin can also lead to a reduction in neurogenesis by inducing apoptosis in proliferating progenitor cells within the hippocampus through the ER stress pathway [25,72]. Recent studies conducted in our laboratory have demonstrated that exposure to deltamethrin increases cleaved caspase-3 levels in cells positive for BrdU, Ki-67, and DCX, indicating the activation of apoptotic signaling in these cells [25]. One key regulator of apoptosis is caspase-12, an ER-resident caspase that is implicated in stress-induced cell death pathways [62]. Under conditions of severe ER stress, caspase-12 is activated and triggers a sequence of downstream signaling events that are ultimately involved in apoptosis, which occurs via upregulation of CHOP [73,74,75]. An increase in caspase-12 expression in BrdU+ and Ki-67+ cells suggests that NPCs may experience ER stress-induced apoptosis after exposure to deltamethrin. We noted a correlation between the decrease in DCX+ cells and the reduction in BrdU- and Ki-67-labeled cells, indicating a potential decline in neurogenesis. The decline in DCX+ immature neurons is likely due to the loss of NSCs (BrdU+ and Ki-67+ cells) via the activation of apoptotic signaling during ER stress, which aligns with earlier findings showing apoptosis of NPCs leading to fewer immature neurons in the hippocampal DGs of adult mice [25]. Indeed, findings from this study demonstrate increase in caspase-12 expression in BrdU+ and Ki-67+ cells, indicating that NPCs may undergo ER stress-induced apoptosis following exposure to deltamethrin.

In summary, we have elucidated a novel mechanism by which exposure to pyrethroid insecticides disrupts hippocampal neurogenesis in mice. Our data demonstrate that NPCs and immature granule neurons undergo ER stress following exposure to deltamethrin. Furthermore, the inhibition of ER stress via salubrinal not only effectively prevents induction of CHOP and ATF4 proteins via deltamethrin but also mitigates the reduction in BrdU-, Ki-67-, and DCX-positive cells in the DG of the hippocampus. Additionally, caspase-12-deficient mice treated with deltamethrin exhibited a dampened ER stress response, resulting in significant protection against the reduction of hippocampal neurogenesis. Collectively, our study elucidates a novel role of ER stress in the disruption of hippocampal neurogenesis after exposure to deltamethrin in mice. The findings further suggest that targeting ER stress pathways could be a potential therapeutic strategy for mitigating the disruption of hippocampal neurogenesis and ameliorating cognitive impairment resulting from pesticide exposure.

## 4. Methods

### 4.1. Chemicals

Deltamethrin (C_22_H_19_Br_2_NO_3_, CAS no. 52918-63-5, purity: 99.5%) was acquired from Chem Service Inc. (West Chester, PA, USA). Ethanol (CH_3_CH_2_OH, CAS no. 64–17–5, purity: 99.5%), acetone (CH_3_COCH_3_, CAS no. 67-64-1, purity: 99.9%), sucrose (C_12_H_22_O_11_, CAS no. 57–50–1, purity: 99.5), HEPES (C_8_H_18_N_2_O_4_S, CAS no. 7365–45–9, purity: 99%), and ethylene glycol (C_2_H_6_O_2_, CAS no.107–21–1, purity: 99%) were obtained from Fisher Scientific (Fair Lawn, NJ, USA). The ER stress inhibitor salubrinal (C_21_H_17_Cl_3_N_4_OS, CAS no. 405060-95-9, purity: 98%), 5-Bromo-2′-Deoxyuridine (C_9_H_11_BrN_2_O_5_, CAS no. 59–14–3, purity: 99%) was purchased from Sigma-Aldrich (St. Louis, MO, USA), and saline solution (0.9% NaCl) was obtained from Hospira (Lake Forest, IL, USA).

### 4.2. Animals

Ten-week-old male C57BL/6j (wild-type) and caspase-12 knockout (KO) mice were obtained from Jackson Laboratories and were housed 5 per cage in an animal care barrier facility with a 12 h light/dark cycle at 22 ± 1 °C and 45 ± 5% relative humidity. Animals were supplied with unlimited access to food (cat #0039980, LabDiet, St. Louis, MO, USA) and water (cat #Hydropac^®^AWS-2500, Seaford, DE, USA). The experiments were carried out in compliance with the NIH Guide for the Care and Use of Laboratory Animals and were approved by the animal care and utilization committee of Florida International University (Approval code: IACUC-21-082-CR01).

### 4.3. Treatment

Sixty wild-type (WT) mice were divided randomly into four groups: control, salubrinal, deltamethrin, and salubrinal + deltamethrin. Additionally, thirty caspase-12 KO mice were divided randomly into two groups: control and deltamethrin. The WT mice received two intraperitoneal (i.p.) injections of 1 mg/kg salubrinal once at 24 h and then 30 min prior to the administration of a single dose of deltamethrin (3 mg/kg) via oral gavage, as shown in Figure 11. Deltamethrin was dissolved in acetone and mixed with corn oil, and then the acetone was removed via evaporation overnight under a biosafety hood. Salubrinal was prepared in physiological saline (0.9% NaCl). The dose of deltamethrin used in this study is 1/20th of the LD50 and close to the lowest observed adverse effect level (LOAEL) of 2.5 mg/kg established by the Environmental Protection Agency (EPA) which induced ER stress and impaired neurogenesis without resulting in any overt toxicity in mice [24,25]. The dosing paradigm for salubrinal was selected based on a prior study, which showed that 1 mg/kg of this compound is highly effective when given 24 h and 30 min before administering an ER stress inducer [64]. The control mice received the same amount of corn oil as the deltamethrin-treated animals. Five animals from each group were sacrificed 48 h after receiving deltamethrin as robust effects observed on ER stress, activation of apoptotic signaling, and reduction in cellular proliferation at this time point in our prior studies [19,25]. Their brains were rapidly removed, and the hippocampi were dissected on ice and stored at −80 °C for biochemical assays.

### 4.4. BrdU Administration and Tissue Preparation

The remaining animals received a single i.p. injection of 300 mg/kg 5-bromo-2-deoxyuridine (BrdU) at 48 h after deltamethrin administration and then 5 animals from each group sacrificed 2 h and 7 day later [25]. Mice were anesthetized with sodium pentobarbital (50 mg/kg, i.p.) and subsequently underwent transcardial perfusion with PBS, followed by 4% paraformaldehyde. The brains were removed, fixed in 4% paraformaldehyde overnight, and then transferred into a PBS solution containing 30% sucrose and 0.1% sodium azide. The brain sections were coronally cut at a thickness of 30 μm using a sliding microtome and then stored in cryoprotectant at −20 °C.

### 4.5. Western Immunoblotting

We conducted Western blot analysis following the methodology outlined in our prior publication [76]. Briefly, hippocampal tissues (*n* = 5/group) were homogenized, and protein concentrations were measured using a BCA (bicinchoninic acid) protein assay kit (cat #23225, Thermo Fisher Scientific, Rockford, IL, USA). Subsequently, 20 µg of protein per lane was separated on 4-12% Bis-Tris Midi gels and transferred onto PVDF membranes. The membranes were then incubated with 7.5% non-fat powder milk for 60 min at room temperature to block non-specific bindings before being incubated overnight at 4 °C with anti-ATF4 (1:1000, Cat#11815, Cell Signaling, Danvers, MA, USA) and anti-CHOP (1:500, cat #SC575; Santa Cruz Biotechnology Inc., Santa Cruz, CA, USA). Following three washes with Tween 20 Tris Buffered Saline (TTBS), suitable horseradish peroxidase-conjugated secondary antibodies were added and incubated for 60 min at room temperature. SuperSignal^®^ West Dura Extended Duration Substrate (Thermo Scientific, Rockford, IL, USA) was used to detect specific antibody-bound proteins with a Bio-Rad imager (Hercules, CA, USA). To ensure equal protein loading, the membranes were then stripped using Pierce Stripping Buffer (Thermo Scientific, Rockford, IL, USA) at room temperature for 15 min and then re-probed with a monoclonal α-tubulin (1:5000, Cat #T5168, Sigma-Aldrich, Inc., St. Louis, MO, USA) or β-actin (1:5000, cat #A5441, Sigma-Aldrich, Inc., St. Louis, MO, USA) antibody. The densitometric value of each band was normalized to the corresponding α-tubulin or β-actin level on the same blot for subsequent statistical analysis.

### 4.6. Immunofluorescence

Immunofluorescent staining was performed following the methodology described in our prior publications [24,25,41]. Every 12th section of the whole hippocampus was utilized. Antigen retrieval was performed by steaming sections in citrate buffer (0.1 M, pH 6.0) for 20 min at 98 °C. To detect BrdU, DNA was denatured by incubating sections in 2 N HCl for 30 min. The sections were treated with 10% normal goat serum (NGS) for 1 h to block nonspecific bindings and then incubated with primary antibodies against BrdU (1:200; cat #5292, Cell Signaling, Danvers, MA, USA), Ki-67 (1:200; cat #550609, BD Pharmingen, San Diego, CA, USA), DCX (1:1000; cat #AB2253, Millipore, MA, USA), caspase-12 (1:500; cat # PIPA587375, Invitrogen, Carlsbad, CA, USA), and CHOP (1:200, cat #AB6199, Abcam, Waltham, MA, USA) overnight at 4 °C. Following three washes with PBS, the sections were incubated with Alexa Fluor 488 and/or 594 dye-conjugated secondary antibodies (Life Technologies, Grand Island, NY, USA) for 1 h at room temperature. Following nuclear staining using DAPI, the sections were mounted onto slides and cover-slipped with VECTASHIELD (cat #H-1400, Vector Laboratories, Newark, CA, USA). To confirm the specificity of staining, a set of sections was incubated without primary antibodies as negative control. BrdU+ and Ki-67+ cells in the subgranular zone (SGZ) and DCX+ cells in the granule cell layer (GCL) were manually counted at high magnification (40×) using an All-in-One Fluorescence Microscope (BZ-X810, Keyence Corporation, Itasca, IL, USA). Quantification was performed on 12 sections/animal from 5 animals/group and was scored by a researcher who was blind to the treatment conditions. Representative images were uniformly adjusted for brightness and contrast using ImageJ 1.53 software to enhance the visualization of immunoreactivity. The group mean was obtained by averaging the counts from these sections, and statistical comparisons were made using two-way ANOVA with Bonferroni’s post hoc test.

### 4.7. Statistical Analysis

Prism 5.01 software (GraphPad Software, San Diego, CA, USA) was used for statistical analysis. The data were presented as mean ± SEM, and one- or two-way ANOVA was performed on the raw data as appropriate. Post hoc multiple comparison tests using Bonferroni’s method were conducted to compare differences among groups after ANOVA. All analyses considered a *p*-value of <0.05 as statistically significant.

## Figures and Tables

**Figure 1 ijms-24-15614-f001:**
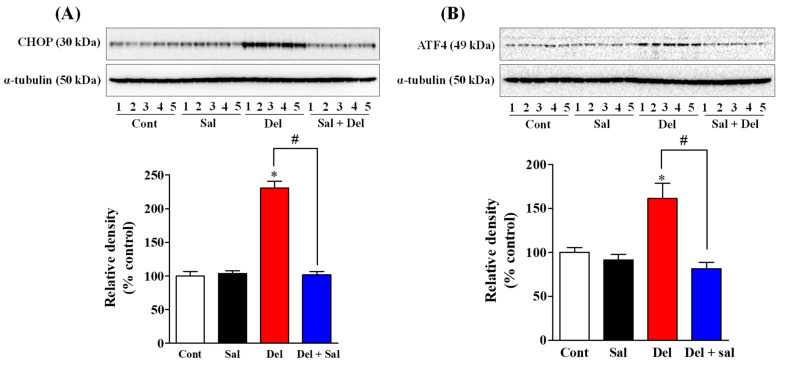
Salubrinal treatment prevents deltamethrin-induced activation of ER stress in the hippocampi of adult mice. Protein levels of ER stress markers CHOP (**A**) and ATF4 (**B**) were determined via Western blot analysis after 48 h of deltamethrin exposure (3 mg/kg, p.o.). Prior to deltamethrin exposure, mice were pre-treated with salubrinal (2 × 1 mg/kg, i.p.) or 0.9% physiological saline solution. Alpha-tubulin served as the housekeeping protein for loading control. All values represent the mean ± SEM from 5 animals per group. An asterisk (*) indicates a significant difference from the control group, and a hashtag (#) indicates a significant difference from the deltamethrin when compared with deltamethrin + salubrinal group (CHOP: *F* 3,16 = 90.58; *p* < 0.001; ATF4: *F* 3,16 = 12.19; *p* < 0.002).

**Figure 2 ijms-24-15614-f002:**
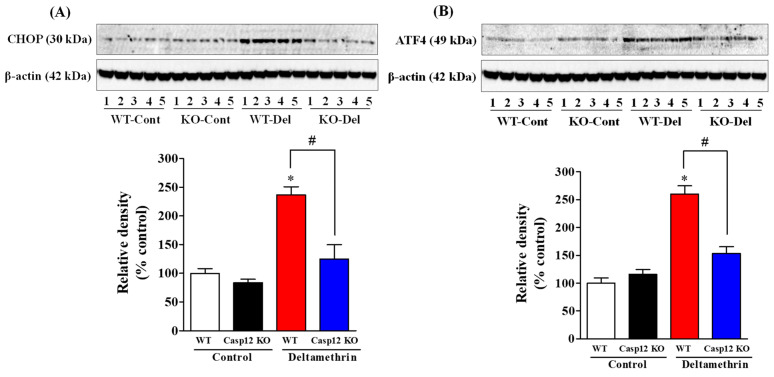
Caspase-12 KO mice exhibit attenuated deltamethrin-induced ER stress in the hippocampus. Protein levels of ER stress markers CHOP (**A**) and ATF4 (**B**) were determined via Western blot analysis after 48 h of deltamethrin exposure (3 mg/kg, p.o.). Beta-actin served as the housekeeping protein for loading control. All values represent the mean ± SEM from 5 animals per group. An asterisk (*) indicates a significant difference from the control group, and a hashtag (#) indicates a significant difference from WT deltamethrin when compared with deltamethrin-treated caspase-12 KO mice (CHOP: *F* 3,16 = 20.21; *p* < 0.001; ATF4: *F* 3,16 = 38.84; *p* < 0.001).

**Figure 3 ijms-24-15614-f003:**
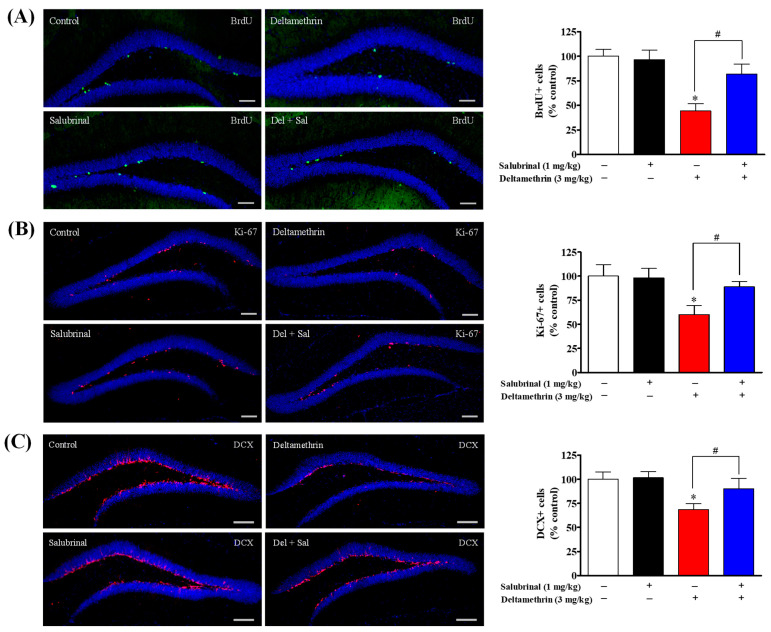
The inhibition of ER stress with salubrinal attenuated the deltamethrin-induced reduction of neurogenesis in the DG of the hippocampi of adult mice. We examined BrdU+ cells (**A**) and Ki-67+ cells (**B**) at 48 h after deltamethrin exposure and DCX+ cells at 7 d after exposure. Immunohistochemistry in the histology section of the dentate gyrus illustrated BrdU+ cells (green), Ki-67+ cells (red), and DCX+ cells (red). Salubrinal treatment attenuated the deltamethrin-induced reduction of BrdU+ cells (**A**), Ki-67+ cells (**B**), and DCX+ cells (**C**). Positive cells in the subgranular zone (SGZ) for BrdU and Ki-67, and in the SGZ and granule cell layer (GCL) for DCX, were manually counted at high magnification (40×) using an All-in-One Fluorescence Microscope (BZ-X810, Keyence Corporation, Itasca, IL, USA). Representative images were uniformly adjusted for brightness and contrast using ImageJ 1.53 software to enhance the visualization of immunoreactivity. Positive cell quantification was performed on 12 sections per animal with 5 animals per group. The group mean was obtained by averaging the counts from these sections, and statistical comparisons were made using two-way ANOVA with Bonferroni’s post hoc test. An asterisk (*) indicates a significant difference from the control group, and a hashtag (#) indicates a significant difference from the deltamethrin when compared with deltamethrin + salubrinal group (BrdU+ cells: *F* 3,12 = 7.29; *p* < 0.048; Ki-67+ cells: *F* 3,16 = 3.77; *p* < 0.032; DCX+ cells: *F* 3,16 = 3.67; *p* < 0.035). Scale bar = 800 µm.

**Figure 4 ijms-24-15614-f004:**
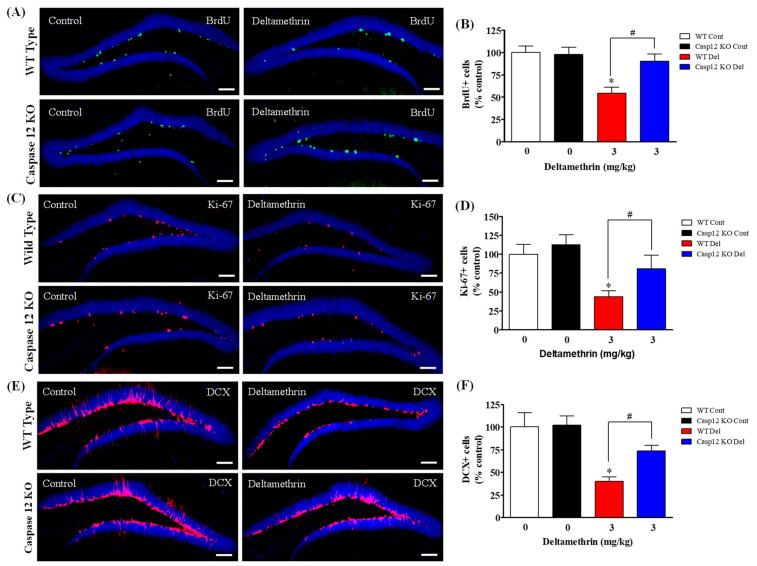
Caspase-12 KO mice exhibit attenuated deltamethrin-induced impairment of neurogenesis in the DG of the hippocampi of adult mice. We examined BrdU+ cells (**A**,**B**) and Ki-67+ cells (**C**,**D**) at 48 h after deltamethrin exposure and DCX+ cells (**E**,**F**) at 7 d after exposure. Immunohistochemistry in the histology section of the dentate gyrus illustrated BrdU+ cells (green), Ki-67+ cells (red), and DCX+ cells (red). Positive cells in the subgranular zone (SGZ) for BrdU and Ki-67, and in the SGZ and granule cell layer (GCL) for DCX were manually counted at high magnification (40×) using an All-in-One Fluorescence Microscope (BZ-X810, Keyence Corporation, Itasca, IL, USA). Representative images were uniformly adjusted for brightness and contrast using ImageJ to enhance the visualization of immunoreactivity. Positive cell quantification was performed on 12 sections per animal with 5 animals per group. The group mean was obtained by averaging the counts from these sections, and statistical comparisons were made using two-way ANOVA with Bonferroni’s post hoc test. An asterisk (*) indicates a significant difference from the control group, and a hashtag (#) indicates a significant difference from the WT deltamethrin when compared with deltamethrin + caspase-12 KO group (BrdU+ cells: *F* 3,12 = 8.81; *p* < 0.0023; Ki-67+ cells: *F* 3,16 = 4.92; *p* < 0.01; DCX+ cells: *F* 3,16 = 8.13; *p* < 0.002). Scale bar = 800 µm.

**Figure 5 ijms-24-15614-f005:**
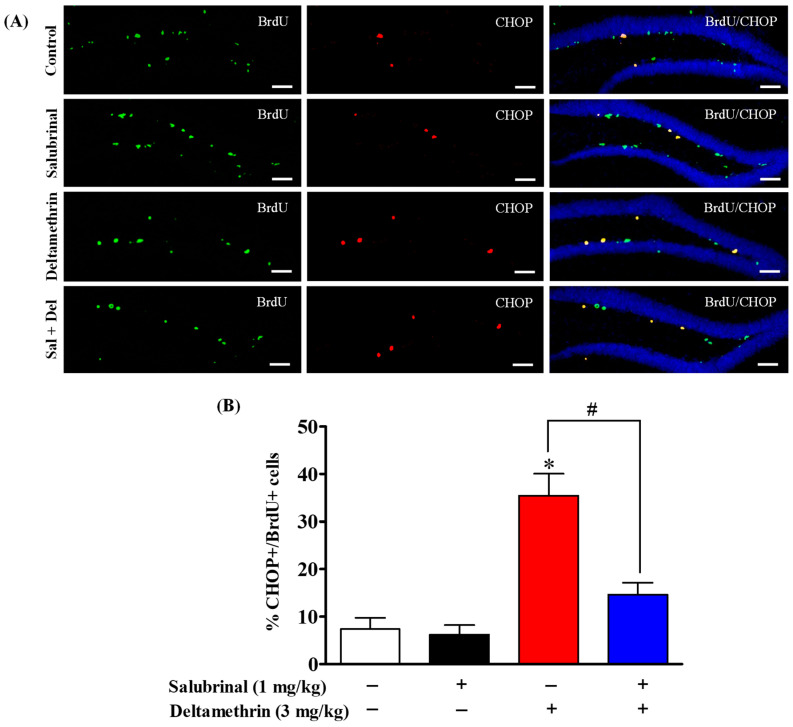
Salubrinal treatment attenuated the deltamethrin-induced increase in the number of CHOP and BrdU co-labeling cells in the DG of the hippocampi of mice. After 48 h of exposure to deltamethrin (3 mg/kg, p.o.), mice were sacrificed for histochemical assessment. (**A**) Representative triple immunofluorescence staining was conducted for BrdU (green), CHOP (red), and DAPI (blue). (**B**) The quantification of BrdU/CHOP-positive cells (yellow) is presented in the bar graph. Representative images were uniformly adjusted for brightness and contrast using ImageJ 1.53 software to enhance the visualization of immunoreactivity. Positive cell quantification was performed on 12 sections per animal with 5 animals per group. The group mean was obtained by averaging the counts from these sections, and statistical comparisons were made using two-way ANOVA with Bonferroni’s post hoc test. An asterisk (*) indicates a significant difference from the control group, and a hashtag (#) indicates a significant difference from the deltamethrin when compared with deltamethrin + salubrinal group (*F* 3,12 = 16.85; *p* < 0.001). Scale bar = 800 μm.

**Figure 6 ijms-24-15614-f006:**
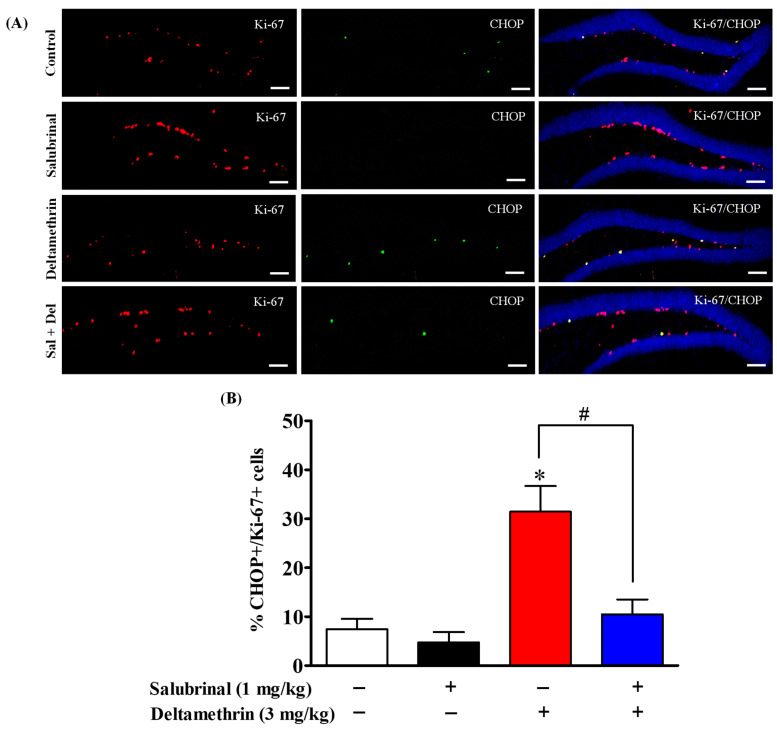
Salubrinal treatment attenuated the deltamethrin-induced increase in the number of CHOP and Ki-67 co-labeling cells in the DG of the hippocampi of adult mice. After 48 h of exposure to deltamethrin (3 mg/kg, p.o.), mice were sacrificed for histochemical assessment. (**A**) Representative triple immunofluorescence staining was conducted for Ki-67 (red), CHOP (green), and DAPI (blue). Positive cells in the subgranular zone (SGZ) were manually counted at high magnification (40×) using an All-in-One Fluorescence Microscope (BZ-X810, Keyence Corporation, Itasca, IL, USA). (**B**) The quantification of Ki-67/CHOP-positive cells (yellow) is presented in the bar graph. Representative images were uniformly adjusted for brightness and contrast using ImageJ 1.53 software to enhance the visualization of immunoreactivity. Positive cell quantification was performed on 12 sections per animal with 5 animals per group. The group mean was obtained by averaging the counts from these sections, and statistical comparisons were made using two-way ANOVA with Bonferroni’s post hoc test. An asterisk (*) indicates a significant difference from the control group, and a hashtag (#) indicates a significant difference from the deltamethrin when compared with the deltamethrin + salubrinal group (*F* 3,12 = 11.99; *p* < 0.005). Scale bar = 800 μm.

**Figure 7 ijms-24-15614-f007:**
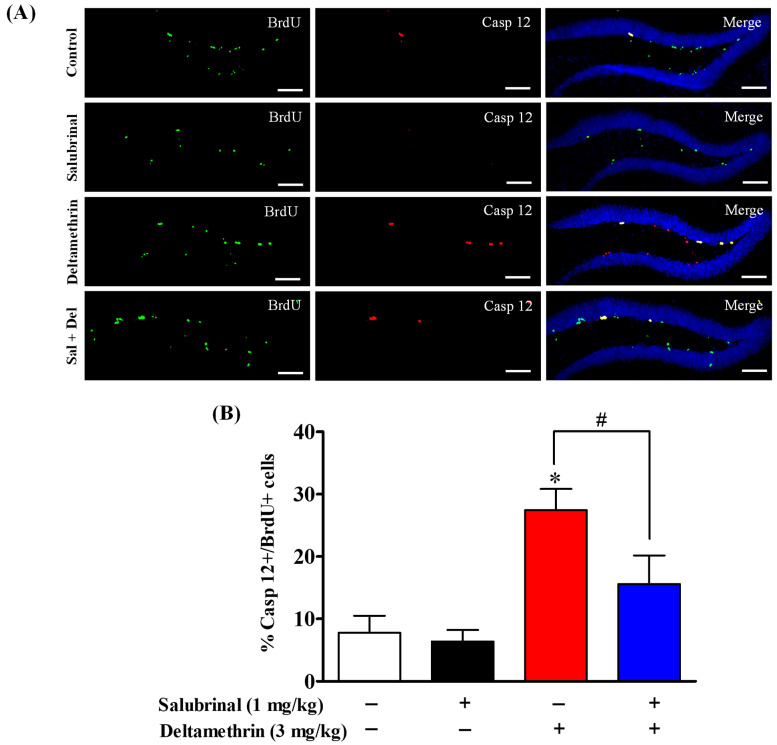
Salubrinal treatment attenuated the deltamethrin-induced increase in the number of caspase-12 and BrdU co-labeling cells in the DG of the hippocampi of mice. After 48 h of exposure to deltamethrin (3 mg/kg, p.o.), mice were sacrificed for histochemical assessment. (**A**) Representative triple immunofluorescence staining was conducted for BrdU (green) and caspase-12 (red), and DAPI (blue). Positive cells in the subgranular zone (SGZ) were manually counted at high magnification (40×) using an All-in-One Fluorescence Microscope (BZ-X810, Keyence Corporation, Itasca, IL, USA). (**B**) Representative images were uniformly adjusted for brightness and contrast using ImageJ 1.53 software to enhance the visualization of immunoreactivity. Positive cell quantification was performed on 12 sections per animal with 5 animals per group. The group mean was obtained by averaging the counts from these sections, and statistical comparisons were made using two-way ANOVA with Bonferroni’s post hoc test. An asterisk (*) indicates a significant difference from the control group, and a hashtag (#) indicates a significant difference from the deltamethrin when compared with deltamethrin + salubrinal group (*F* 3,12 = 10.20; *p* < 0.001). Scale bar = 800 μm.

**Figure 8 ijms-24-15614-f008:**
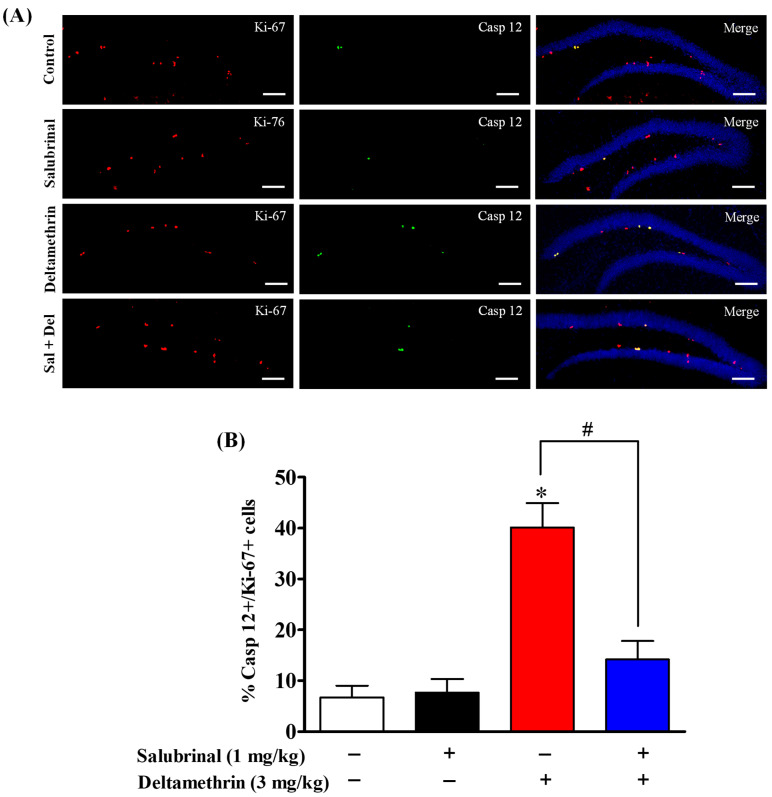
Salubrinal treatment attenuated the deltamethrin-induced increase in the number of caspase-12 and Ki-67 co-labeling cells in the DG of the hippocampi of adult mice. After 48 h of exposure to deltamethrin (3 mg/kg, p.o.), mice were sacrificed for histochemical assessment. (**A**) Representative triple immunofluorescence staining was conducted for Ki-67 (red) and caspase-12 (green), and DAPI (blue). Positive cells in the subgranular zone (SGZ) were manually counted at high magnification (40×) using an All-in-One Fluorescence Microscope (BZ-X810, Keyence Corporation, Itasca, IL, USA). (**B**) The quantification of caspase-12/Ki-67+ cells (yellow) are presented in the bar graph. Positive cell quantification was performed on 12 sections per animal with 5 animals per group. The group mean was obtained by averaging the counts from these sections, and statistical comparisons were made using two-way ANOVA with Bonferroni’s post hoc test. An asterisk (*) indicates a significant difference from the control group, and a hashtag (#) indicates a significant difference from the deltamethrin when compared with deltamethrin + salubrinal group (*F* 3,12 = 34.96; *p* < 0.001). Scale bar = 800 μm.

**Figure 9 ijms-24-15614-f009:**
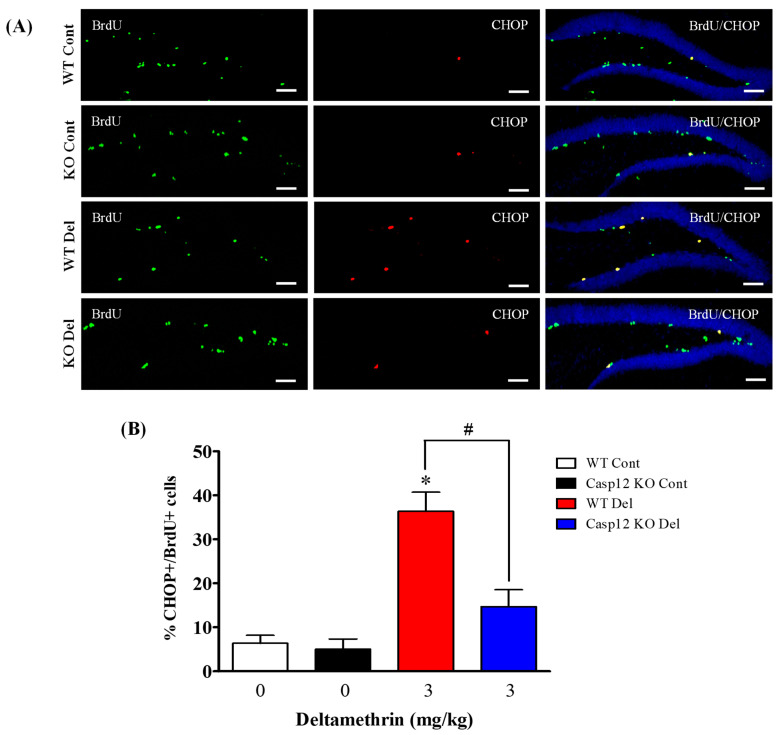
Caspase-12 KO mice exhibit attenuated the deltamethrin-induced increase in the number of CHOP and BrdU co-labeling cells in the DG of the hippocampi of adult mice. After 48 h of exposure to deltamethrin (3 mg/kg, p.o.), mice were sacrificed for histochemical assessment. (**A**) Representative triple immunofluorescence staining was conducted for BrdU (green), CHOP (red), and DAPI (blue). Positive cells in the subgranular zone (SGZ) were manually counted at high magnification (40×) using an All-in-One Fluorescence Microscope (BZ-X810, Keyence Corporation, Itasca, IL, USA). (**B**) The quantification of BrdU/CHOP-positive cells (yellow) is presented in the bar graph. Representative images were uniformly adjusted for brightness and contrast using ImageJ to enhance the visualization of immunoreactivity. Positive cell quantification was performed on 12 sections per animal from 5 animals per group. The group mean was obtained by averaging the counts from these sections, and statistical comparisons were made using two-way ANOVA with Bonferroni’s post hoc test. An asterisk (*) indicates a significant difference from the control group, and a hashtag (#) indicates a significant difference from the deltamethrin when compared with deltamethrin + caspase-12 KO group (*F* 3,12 = 18.21; *p* < 0.001). Scale bar = 800 μm.

**Figure 10 ijms-24-15614-f010:**
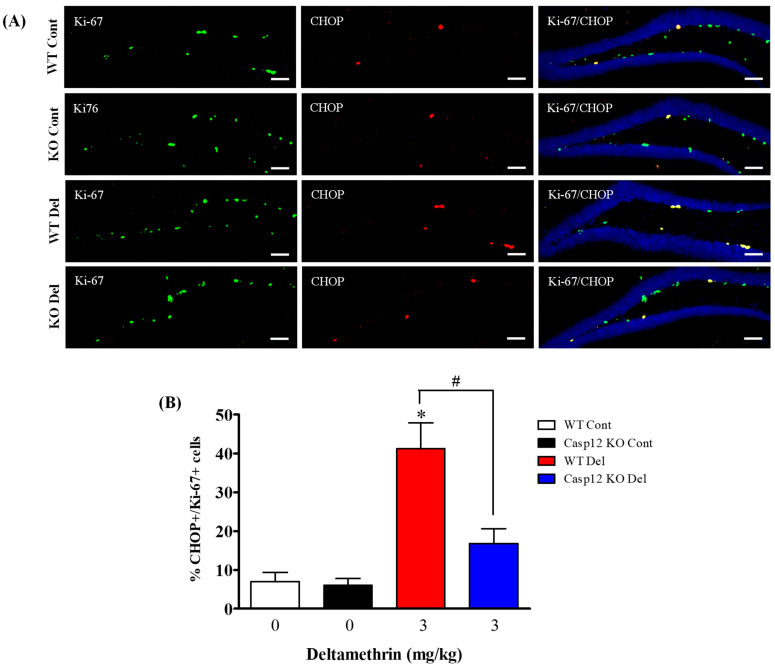
Caspase-12 KO mice exhibit attenuated the deltamethrin-induced increase in the number of CHOP and Ki-67 co-labeling cells in the DG of the hippocampi of mice. After 48 h of exposure to deltamethrin (3 mg/kg, p.o.), mice were sacrificed for histochemical assessment. (**A**) Representative triple immunofluorescence staining was conducted for Ki-67 (green), CHOP (red), and DAPI (blue). (**B**) Positive cells in the subgranular zone (SGZ) were manually counted at high magnification (40×) using an All-in-One Fluorescence Microscope (BZ-X810, Keyence Corporation, Itasca, IL, USA). The quantification of Ki-67/CHOP-positive cells (yellow) is presented in the bar graph. Representative images were uniformly adjusted for brightness and contrast using ImageJ 1.53 software to enhance the visualization of immunoreactivity. Positive cell quantification was performed on 12 sections per animal from 5 animals per group. The group mean was obtained by averaging the counts from these sections, and statistical comparisons were made using two-way ANOVA with Bonferroni’s post hoc test. An asterisk (*) indicates a significant difference from the control group, and a hashtag (#) indicates a significant difference from deltamethrin when compared with deltamethrin + caspase-12 KO group (*F* 3,12 = 16.01; *p* < 0.002). Scale bar = 800 μm.

**Figure 11 ijms-24-15614-f011:**
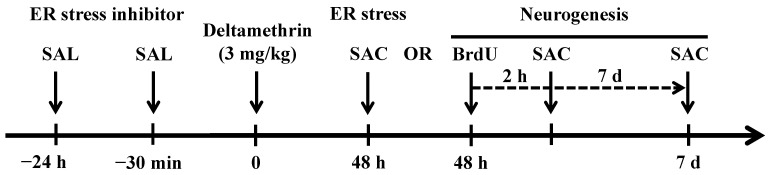
The experimental paradigm and timeline for investigating the role of ER stress on adult hippocampal neurogenesis. Mice were pre-treated with salubrinal (1 mg/kg, i.p.) and then given either 0 (corn oil) or 3 mg/kg of deltamethrin via oral gavage. Following 48 h of deltamethrin exposure, a set of animals was sacrificed for ER stress assay. Another set of animals received BrdU (300 mg/kg, i.p.) and were then sacrificed 2 h and 7 days later for neurogenesis assays.

## Data Availability

The data presented in this study are available in the article and Appendix A.

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
