# Peer review of "The Preventive Effects of Salubrinal against Pyrethroid-Induced Disruption of Adult Hippocampal Neurogenesis in Mice"

_ijms, 2023, doi:10.3390/ijms242115614_

Round 1

Reviewer 1 Report

The Article titled "Pyrethroid exposure causes ER stress-mediated disruption of adult hippocampal neurogenesis in mice" is an interesting work, but major information is missing to conclude.

The Article is well-organized, presenting promising results. 

The title didn’t reflect the totality of the idea of the article (I suggest adding the preventive effect of Salubrinal in the title)

In previous studies, you used 3mg and 6mg of deltamethrin via oral gavage. Why did you choose 3mg for this study?

In the previous study, you have used 24h and 48h treatment periods. Why did you choose 48 for this study?

The author needs to explain why the mice received two intraperitoneal injections of 1 mg/kg salubrinal at once at 24 h and then 30 min before administering a single dose of deltamethrin (3 mg/kg).

How you have chosen two injections and why at this specific time (24 h and then 30 min before the administration of a single dose of deltamethrin)

The author needs to include the signaling pathway regulation to explain more the effects in the discussion.

It is also crucial to add a scientific explanation of the preventive effect of Salubrinal.

  Additionally, it will be essential to evaluate the RNA expression of the key genes in this process.

The author must show the major revisions made in the text by highlighting the changes in a different colored text. 

It is imperative to consider all these remarks to reinforce the manuscript's quality and conclude more accurately.

Author Response

We greatly appreciate the reviewer's thoughtful comments and suggestions, which helped us improve the quality of our manuscript. All the corrections, additions, and modifications are in red and highlighted in yellow in our revised manuscript.

The title didn’t reflect the totality of the idea of the article (I suggest adding the preventive effect of Salubrinal in the title)

Response: We have revised the title as “The preventive effects of salubrinal against pyrethroid-induced disruption of adult hippocampal neurogenesis in mice” according to your suggestion to reflect the idea of the article.

In previous studies, you used 3mg and 6mg of deltamethrin via oral gavage. Why did you choose 3mg for this study?

Response: In our previous study, we used a 6 mg/kg dose for an acute examination to explore the susceptibility of brain regions to ER stress and apoptosis following exposure to deltamethrin. The dose of 6 mg/kg of deltamethrin is 1/10th of the LD50 (Sharma et al., 2014) and did not cause any classic signs of toxicity, including tremors, salivation, choreoathetosis, and ataxia in mice. Given the magnitude of the effects at 6 mg/kg of deltamethrin, we next examined whether similar effects could be observed with a lower dose. We used 3 mg/kg of deltamethrin which is 1/20th of the LD50 and is close to the lowest observed adverse effect level (LOAEL) of 2.5 mg/kg established by the EPA (2010) in the risk assessment of deltamethrin. Our findings demonstrated that 3 mg/kg of deltamethrin also induced ER stress and influenced neurogenesis without resulting in any overt toxicity. Based on these observations, we extended our study with the 3 mg/kg dose to elucidate the mechanisms underlying the disruption of hippocampal neurogenesis in this current study.

In the previous study, you have used 24h and 48h treatment periods. Why did you choose 48 for this study?

Response: For this study, we selected a 48 h treatment period based on the robust effects observed on ER stress, activation of apoptotic signaling, and reduction in cellular proliferation in our prior studies following deltamethrin administration (Hossain et at., 2019, 2022).

The author needs to explain why the mice received two intraperitoneal injections of 1 mg/kg salubrinal at once at 24 h and then 30 min before administering a single dose of deltamethrin (3 mg/kg).

Response: Mice were given two intraperitoneal (i.p.) injections of 1 mg/kg salubrinal at 24 h and 30 min before the administration of deltamethrin. This dosing paradigm was chosen based on a prior study which demonstrated that pretreating mice with two doses of salubrinal (1 mg/kg, i.p.) is highly effective in preventing kainic acid-induced ER stress when given 24 h and 30 min prior to kainic acid exposure (Kim et al., 2014).

How you have chosen two injections and why at this specific time (24 h and then 30 min before the administration of a single dose of deltamethrin)

Response: This 24 h allows for sufficient time for the compound to be absorbed, distributed, metabolized, and reach the target cells or tissues where it exerts its protective effects against ER stress. The second dose at 30 min before the introduction of deltamethrin to ensures a substantial concentration of salubrinal in the system precisely when it is needed to counteract the potential ER stress induced by deltamethrin.

The author needs to include the signaling pathway regulation to explain more the effects in the discussion.

Response: Signaling pathways has been discussed in the discussion section. During ER stress, cells trigger unfolded protein response (UPR) to restore ER functions through three ER-transmembrane sensors, namely protein kinase RNA-like endoplasmic reticulum kinase (PERK), inositol-requiring enzyme 1 (IRE1), and transcription factor 6 (ATF6). Upon the activation of the UPR in ER, eIF2alpha undergoes phosphorylation through PERK pathways. This event triggers the translational induction of ATF4, which subsequently activates CHOP, ultimately leading to the propagation of pro-apoptotic signals. Dysregulation of this signaling cascade has been implicated in neurodegeneration and neurodegenerative diseases. Our data demonstrates that inhibiting eIF2α dephosphorylation with salubrinal prevents ATF4 and CHOP induction in the hippocampus and mitigates their expression in NPCs in deltamethrin-treated mice.

It is also crucial to add a scientific explanation of the preventive effect of Salubrinal.

Response: Scientific explanation of preventive effects of salubrinal has been provided in discussion section. Salubrinal is a specific inhibitor of eIF2-α dephosphorylation, which serves as an initial signaling molecule in ER stress. By preventing this dephosphorylation, salubrinal helps to reduce the burden on the ER and restore cellular homeostasis.

Additionally, it will be essential to evaluate the RNA expression of the key genes in this process.

Response: Given the fact that mRNA expression serves as the intermediate step in protein synthesis and proteins are the functional executors in cellular processes, we directly measured ER stress-related CHOP and ATF4 protein levels without requiring mRNA expression analysis in this study.

Reviewer 2 Report

The study by Toltin et al. report that eukaryotic initiation factor-2α (eIF2α) inhibitor, salubrinal or knockout of caspase-12 attenuated the elevated expression of C/EBP Homologous Protein (CHOP) transcription factor and activating transcription factor 4 (ATF4) in the hippocampus after deltamethrin treatment. Further, the authors suggest that the salubrinal or knockout of caspase-12 restored the impaired neurogenesis caused by deltamethrin. The functional role of ER stress in altered neurogenesis by pesticide toxicity is significant to understand the mechanism of the neurological diseases and its treatment strategy. However, there are several concerns which should be addressed by the authors. Some specific comments to consider are suggested below.

A detailed timeline diagram for each experiment would help to understand the authors' procedure.

The authors need to provide sufficient information for the experiments in the manuscript. For example (line 115), the detailed procedure was not described in reference 45.

It is important to provide details about the cell counting methodology. Since so much of this paper hinges on cell count data, it is critical that the methodology is described in detail. How did the authors decide the area for counting and normalize? In most of the figures it needs to be clarified how imaging parameters, particularly signal thresholds, were defined to discriminate between signal positive and negative cells. The statistics for defining double-positive or negative cells needs to be explained in greater detail. Since % is shown, what was the total number of cells in each case?  Was this number comparable across conditions? Maybe add a table to Figures containing all the cell counts and percent calculations for all experimental groups.

Species of antibodies need to be indicated. To make sure the reproducibility, please provide Research Resource Identifiers (RRIDs) for all animals and relevant materials (chemicals, antibodies etc.) in the Materials and Methods section.

Some definitions of abbreviations are missing or incomplete.

There is a lack of consistency. eIf2α or eIF2-α need to be selected.

Line 143 and 155: Sentences are incomplete.

How is the expression of three major ER stress sensors, such as inositol-requiring enzyme 1 (IRE1), transcription factor 6 (ATF6) and protein kinase RNA-like endoplasmic reticulum kinase (PERK) after treatment with deltamethrin and/or salubrinal?

How is the total and activated caspase-12 expression in neural stem/progenitor cells after 3 mg/kg deltamethrin treatment? If altered, could salubrinal treatment restore the levels of their expression?

Was BrdU+ counted within overlapping DAPI expression? Need to describe the detailed methodology of imaging analyses.

Figure 4: Very low Ki-67 expression in control of caspase-KO mice is shown in Figure 4C. However, the column (black bar) shows contradictory in Figure 4D.

Figures 5 to 8: The description of y-axis is confused. For example, "% of CHOP+ in BrdU + cells (Figure 5B)" or "% of CHOP+/BrdU+ cells (Figure 5B)"would be easy to understand.

Were Figure 5 and Figure 6 the different timelines? If same timeline, what is the significant difference of those figures. There is the same question about Figures 7 and 8.

Each figure legend needs to provide sufficient information for the readers. Legends must fully explain the figure without reference to the text.

The postnatal neural stem cell (NSC) pool contains quiescent and activated NSCs. The authors showed the decreased numbers of BrdU+/Ki-67+ and DCX+ cells after deltamethrin treatment. It is not clear if the decrease is because of the increase/decrease of NSC activation or the decrease of NSC numbers. To investigate Sox2 expression may help the authors' conclusion.

Some definitions of abbreviations are missing or incomplete.

Line 143 and 155: Sentences are incomplete.

Author Response

We greatly appreciate the reviewer's thoughtful comments and suggestions, which helped us improve the quality of our manuscript. All the corrections, additions, and modifications are in red and highlighted in yellow in our revised manuscript.

A detailed timeline diagram for each experiment would help to understand the authors' procedure.

Response: We have included the experimental paradigm and timeline for investigating the role of ER stress on adult hippocampal neurogenesis in our revised manuscript (refer to Figure 1).

The authors need to provide sufficient information for the experiments in the manuscript. For example (line 115), the detailed procedure was not described in reference 45.

Response: We have provided sufficient detail for the experimental procedures, including the western blot analysis in our revised manuscript.

It is important to provide details about the cell counting methodology. Since so much of this paper hinges on cell count data, it is critical that the methodology is described in detail. How did the authors decide the area for counting and normalize? In most of the figures it needs to be clarified how imaging parameters, particularly signal thresholds, were defined to discriminate between signal positive and negative cells. The statistics for defining double-positive or negative cells needs to be explained in greater detail. Since % is shown, what was the total number of cells in each case?  Was this number comparable across conditions? Maybe add a table to Figures containing all the cell counts and percent calculations for all experimental groups.

Response: We have provided details on the cell counting methods and other criteria considered for neurogenesis assessment, as well as the statistical analysis. The total number of cells in each case has been provides in tables (refer to attached Table 1 and Table 2)

Table1.  Total BrdU+, Ki-67+, DCX+, CHOP+, and caspase-12+ cells in the DG of the hippocampus following acute deltamethrin exposure in mice.

Cell Type

Control

Salubrinal

Deltamethrin

Salubrinal + Deltamethrin

BrdU+ cells

183

177

81**

150*

Ki-67+ cells

245

240

147**

218*

DCX+ cells

1130

1147

775**

1017*

CHOP+/BrdU+ cells

15

13

56**

22*

CHOP+/Ki67+ cells

14

12

60**

24*

Casp 12+/BrdU+ cells

15

12

66**

30*

Casp 12+/Ki67+ cells

16

15

69**

27*

Total BrdU+, Ki-67+, DCX+, and CHOP+ cells were visualized through immunofluorescence. Positive cells in the subgranular zone (SGZ) for BrdU and Ki-67, and in the SGZ and granule cell layer (GCL) for DCX, were manually counted at high magnification (40X) using an All-in-One Fluorescence Microscope (BZ-X810, Keyence Corporation, Itasca, IL). Positive cell quantification was performed on 12 sections per animal with 5 animals in each group. The group mean was obtained by averaging the total counts from these sections. Statistical comparisons were made using two-way ANOVA with Bonferroni’s post hoc test. An asterisk (**) indicates a significant difference from the control group and a hashtag (*) indicates a significant difference from deltamethrin when compared with the deltamethrin + salubrinal group (BrdU+ cells: F 3,12 = 7.29; p < 0.048; Ki-67+ cells: F 3,16 = 3.77; p < 0.032; DCX+ cells: F 3,16 = 3.67; p < 0.035; CHOP+/BrdU+: F 3,12 = 16.85; p < 0.001; CHOP+/Ki-67+ cells: F 3,12= 11.99; p < 0.005; Caspase 12+/BrdU+ cells: F 3,12 = 10.20; p < 0.001; Caspase 12+/Ki67+ cells: F 3,12 = 34.96; p < 0.001).

Table 2. Total BrdU+, Ki-67+, DCX+ and CHOP+ cells in the DG of the hippocampus of caspase-12 KO mice following acute exposure to deltamethrin.

Cell Type

WT Cont

Caspase 12 KO Cont

WT Del

Caspase 12 KO + Del

BrdU+ cells

171

167

 93**

153*

Ki-67+ cells

255

287

112* 

206*

DCX+ cells

1209

1231

486**

891*

CHOP+/BrdU+ cells

12

9

56**

22*

CHOP+/Ki67+ cells

11

10

62**

30*

Total BrdU+, Ki-67+, DCX+, and CHOP+ cells were visualized through immunofluorescence. Positive cells in the subgranular zone (SGZ) for BrdU and Ki-67, and in the SGZ and granule cell layer (GCL) for DCX, were manually counted at high magnification (40X) using an All-in-One Fluorescence Microscope (BZ-X810, Keyence Corporation, Itasca, IL). Positive cell quantification was performed on 12 sections per animal with 5 animals in each group. The group mean was obtained by averaging the total counts from these sections. Statistical comparisons were made using two-way ANOVA with Bonferroni’s post hoc test. An asterisk (**) indicates a significant difference from the control group and a hashtag (*) indicates a significant difference from deltamethrin when compared with the deltamethrin + salubrinal group (BrdU+ cells: F 3,12 = 8.81; p < 0.0023; Ki-67+ cells: F 3,16 = 4.92; p < 0.01; DCX+ cells: F 3,16 = 8.13; p < 0.002; CHOP+/BrdU+ cells: F 3,12 = 18.21; p < 0.001; CHOP+/Ki67+ cells: F 3,12 = 16.01; p < 0.002)

Species of antibodies need to be indicated. To make sure the reproducibility, please provide Research Resource Identifiers (RRIDs) for all animals and relevant materials (chemicals, antibodies etc.) in the Materials and Methods section.

Response: Sources of relevant chemicals and antibodies, along with their catalog numbers, have been included in the methods section of our revised manuscript for the purpose of reproducibility. Additionally, information regarding the sources of animals has also been provided.

Some definitions of abbreviations are missing or incomplete.

Response: We have thoroughly addressed the definitions of abbreviations in our revised manuscript.

 There is a lack of consistency. eIf2α or eIF2-α need to be selected.

 Response: We have revised manuscript accordingly with eIf2α

Line 143 and 155: Sentences are incomplete.

Response: Sentences have been corrected (line 143 is now line 177 and line 155 is now line 191) in our revised manuscript.

How is the expression of three major ER stress sensors, such as inositol-requiring enzyme 1 (IRE1), transcription factor 6 (ATF6) and protein kinase RNA-like endoplasmic reticulum kinase (PERK) after treatment with deltamethrin and/or salubrinal?

Response: In this study, our focus was on the downstream effects of the PERK pathway through the inhibition of eIF2α using salubrinal. Due to a shortage of time and unavailability of antibodies, we regret to inform you that we were unable to investigate the expression of ER stress sensors at this time. However, we have plans to address this in our future studies.

How is the total and activated caspase-12 expression in neural stem/progenitor cells after 3 mg/kg deltamethrin treatment? If altered, could salubrinal treatment restore the levels of their expression?

Response: According to your suggestion, we conducted additional experiments to examine the activation of caspase-12 expression in neuronal progenitor cells and have included this data in our revised manuscript (Figures 8 and 9). Yes, salubrinal significantly attenuated the number of caspase-12 expressing neuronal progenitor cells.

Was BrdU+ counted within overlapping DAPI expression? Need to describe the detailed methodology of imaging analyses.

Response: No, BrdU counting was not overlap with DAPI expression. We have described detailed methodology of imaging analysis in our revised manuscript. No, there was no overlap between BrdU counting and DAPI expression. We have provided a detailed description of the imaging analysis methodology in our revised manuscript.

Figure 4: Very low Ki-67 expression in control of caspase-KO mice is shown in Figure 4C. However, the column (black bar) shows contradictory in Figure 4D.

Response: Our apologies for this mistake. We have revised figure 4C with correct image for Ki67 that now reflect respective black bar in figure 4D.

Figures 5 to 8: The description of y-axis is confused. For example, "% of CHOP+ in BrdU + cells (Figure 5B)" or "% of CHOP+/BrdU+ cells (Figure 5B)"would be easy to understand.

Response: As per your suggestion, Figures 5 to 8 have been revised. The X-axis now represents “% of CHOP+/BrdU+ cells”.

Were Figure 5 and Figure 6 the different timelines? If same timeline, what is the significant difference of those figures. There is the same question about Figures 7 and 8.

Response: We have included time points in the figure legends, and the justification has been provided in the results section (refer to lines 287-290). We examined BrdU+ cells (A) and Ki-67+ cells (B) at 48 h after deltamethrin exposure and DCX+ cells at 7 d after exposure as no changes was observed at 48 h in our previous study (Hossain etl., 2022). We have refined the figure legends to enhance the clarity regarding significant differences in these figures.

Each figure legend needs to provide sufficient information for the readers. Legends must fully explain the figure without reference to the text.

Response: All figure legends have been updated with sufficient information for ensuring that they comprehensively describe the figures without the need for reference to the main text.

The postnatal neural stem cell (NSC) pool contains quiescent and activated NSCs. The authors showed the decreased numbers of BrdU+/Ki-67+ and DCX+ cells after deltamethrin treatment. It is not clear if the decrease is because of the increase/decrease of NSC activation or the decrease of NSC numbers. To investigate Sox2 expression may help the authors' conclusion.

Response: We have discussed this matter in our revised manuscript (lines 545-551). The decline in DCX+ immature neurons is likely due to the loss of NSCs (BrdU+ and Ki-67+ cells) through the activation of apoptotic signaling during ER stress, which aligns with earlier findings showing apoptosis of NPCs leading to fewer immature neurons in the hippocampal DG of adult mice [26]. Indeed, findings from this study demonstrate increase in caspase-12 expression in BrdU+ and Ki-67+ cells, indicating that NPCs may undergo ER stress-induced apoptosis following exposure to deltamethrin.

Round 2

Reviewer 1 Report

In this version of the article, “The preventive effects of salubrinal against pyrethroid-induced disruption of adult hippocampal neurogenesis in mice.” We can see an acceptable evolution compared to the first version because it has become more structured with more explanation.

The author has clarified many points in the treatment design chosen and responded to the questions.

And also, the author has enriched the discussion with the regulation pathway. 

Furthermore, the author has considered the reviewer's remarks and suggestions, which has positively impacted the quality and consistency of the article.

with this version, the article shows an excellent scientific level and represents an added value in the interested research topics 

the article is accepted for me with this version